# PN Codes Estimation of Binary Phase Shift Keying Signal Based on Sparse Recovery for Radar Jammer

**DOI:** 10.3390/s23010554

**Published:** 2023-01-03

**Authors:** Bo Peng, Qile Chen

**Affiliations:** School of Mechatronical Engineering, Beijing Institute of Technology, Beijing 100089, China

**Keywords:** radar jammer, parameter estimation, binary phase shift keying, PN codes, nonconvex total variation regularization

## Abstract

Parameter estimation is extremely important for a radar jammer. With binary phase shift keying (BPSK) signals widely applied in radar systems, estimating the parameters of BPSK signals has attracted increasing attention. However, the BPSK signal is difficult to be processed by traditional time frequency analysis methods due to its phase jumping and abrupt discontinuity features which makes it difficult to extract PN (PN) codes of the BPSK signal. To solve this problem, a two-step PN codes estimation method based on sparse recovery is introduced in this paper. The proposed method first pretreats the BPSK signal by estimating its center frequency and converting it to zero intermediate frequency (ZIF). The pretreatment transforms phase jumps of the BPSK signal into the level jumps of the ZIF signal. By nonconvex sparsity promoting regularization, the level jumps of the ZIF signal are extracted through an iterative algorithm. Its effectiveness is verified by numeric simulations and semiphysical tests. The corresponding results demonstrate that the proposed method is able to estimate PN codes from the BPSK signal in serious electromagnetic environments.

## 1. Introduction

With the development of electronic warfare, the radar jammer has become one of the most important equipment of the modern war [1]. It senses the hostile radars and invalidates them by radioing jamming signal to them [2,3]. The radar jammer is able to estimate the parameters of the radar signal and generate a coherent jamming signal based on the estimated results. Parameter estimation of the radar signal is extremely important for radar jammer which provide guidance for radar jamming [4,5]. A large number of parameter estimation methods for multiple types of radar signals have been proposed [6,7,8,9,10,11,12].

Possessing low probability of interception and strong anti-interference ability [13,14,15], binary phase shift keying (BPSK) signals are widely applied in radar. Therefore, parameter estimation of BPSK signals for radar jammer has attracted great attention [16,17]. Various time frequency analysis methods, such as fast Fourier transform [18], short time Fourier transform [19], time–frequency distribution based on Ville–Wigner distribution [20], wavelet transform [21] and cyclostationary [10,11,12] have been used to extract the parameter of the BPSK signal. However, these methods only focus on estimating the carrier frequency or chip rate of the BPSK signal and fail to extract its PN codes.

Recently, some methods for PN codes estimation of the BPSK signal have been proposed. By synchronous demodulation, ref. [22] succeeds to extract the PN codes of a BPSK signal. However, this method shows poor performance in serious environment. It needs several priors, such as the chip rate and the center frequency of the signal. Ref. [23] adopts the matrix eigen decomposition to estimate the PN codes of the BPSK signal, which performs competitive in low signal to noise ratio (SNR). However, the period of the PN codes and its chip rate must be known when adopting this method. Ref. [16] proposes a PN codes estimation method based on duffing oscillator. It detects the polarity changes of the PN codes according to the state changes of the duffing oscillator. As the insensitivity of the duffing oscillator to noise, this method performs well even in serous SNR. However, it only detects the polarity changes of the PN codes and need the starting symbol as a prior. Ref. [24] proposes a two-stage method based on cross-correlation function which estimates the carrier frequency of the BPSK signal at first stage and estimates its the PN codes at second stage. However, this method is only suitable for Barker codes 7, 11 and 13.

The difficulty of PN codes estimation lies in the lack of appropriate signal analysis methods. The phase jumping and abrupt discontinuity features of BPSK signals lead the instantaneous frequency to an impulse function that is present at all frequencies. The current signal analysis methods, such as the short-time Fourier transform, Wigner–Ville distribution [25], wavelet transform [26], empirical mode decomposition [27], and variation modal decomposition [28], are defined on orthogonal basis functions or intrinsic mode function and suffer from dealing with phase jumping and abrupt discontinuity features. Recently, nonlinear filtering has attracted a great deal of attention in the field of feature extraction which are able to extract the jump discontinuities of a signal based on its sparsity [29,30,31]. The sparsity of a signal is always constrained by L0 norm [32]. However, the L0 norm is a NP-hard problem. Therefore, other norms, such as L1 norm, are used to replace the L0 norm to alleviate this problem. The L1 norm regularization, which is called LASSO, achieves excellent performance for its strongly convex and easy to be solved. However, it tends to underestimate the amplitude of the signal discontinuities and has poor tolerance to noise. Therefore, some nonconvex penalty functions are explored instead of the L1 norm to overcome the shortages of the L1 norm [33,34,35]. The minimax concave (MC) penalty [36], considered in our sparse optimization, falls in this nonconvex sparsity-inducing penalty class.

Inspired by the previous works, a novel two-step PN codeS estimation method based on nonconvex total variation regularization (NCTVR) is introduced for BPSK signal. The NCTVR-based method first pretreats the BPSK signal by estimating its center frequency with an interpolation to Fourier coefficients (IFCs) frequency estimator [37] and converting it to ZIF. Then, an NCTVR filter is adopted to extract the PN codes from the ZIF signal. Different from the L1 norm-based regularization, we introduce a MC function as the penalty to promote the strong sparsity of the PN codes which also improves its tolerance to noise. An iterative algorithm based on the forward–backward splitting algorithm is proposed to solve the NCTVR optimization problem. Finally, we verified the effectiveness of the NCTVR-based method by numeric simulations and semiphysical tests. The main contributions of this paper are as follows:
A novel PN codes estimation method based on NCTVR is proposed, and its corresponding optimization function is established.An iterative algorithm based on the forward–backward splitting algorithm is proposed to solve the NCTVR.The proposed method is verified by numeric simulations and semiphysical tests.

To make the novelty of the proposed method more clearly, the difference among our method and the previous works are given in Table 1.

The remainder of this paper is organized as follows. The mathematical model is given in Section 2. The principle of the NCTVR-based method proposed in this paper and a detailed outline of the iterative algorithm are presented in Section 3. In Section 4, the essence and performance of the NCTVR-based method are described. In Section 5, simulations and semiphysical tests are shown. Conclusions are formed in Section 6.

## 2. Mathematical Model

In modern electronic warfare, the radar jammer is widely applied for protecting owned units from the detection or guidance of the hostile radar systems. The principle of the radar jammer is shown in Figure 1. When capturing a radar signal, the radar jammer first converts it to digital intermediate frequency signal with the local oscillator and high-speed A/Ds. Then, the parameters of the radar signal are estimated from the digital intermediate frequency signal. Based on the parameters of the radar signal, a jamming signal is generated and upconverted to radio frequency. By accurate parameters estimation, the radar jammer is able to invalidate the hostile radar.

When the receiver of the jammer captures a radar signal, it first quadrature downconverts the captured signal into intermediate frequency with a local oscillator. Then, a pair of high-speed analog-to-digital converters (ADCs) with sampling rate $f_{s}$ is used to convert the analogue intermediate frequency signal to a digital intermediate frequency signal. Assuming the captured signal is a BPSK signal, and the sampling interval of the ADCs is $T_{s}=\frac{1}{f_{s}}$, the digital intermediate frequency signal obtained from sampling can be expressed as
(1)$$\begin{array}{r} s_{dif}(n)=\sum_{k=0}^{K-1}c_{k}rect(n-k\frac{T_{c}}{T_{s}})e^{j2\pi f_{id}n+\varphi_{0}}+w(n),\\ n=1,2\cdots N \end{array}$$
where $c_{k}=\left\{+1,-1\right\}$ denote PN codes with length K, $T_{c}$ is the chipping width, N is the number of samples, $rect(n)=\{\begin{array}{c} 1,0<n<\frac{T_{c}}{T_{s}}\\ 0,else \end{array}$ is a rectangular window function, $j^{2}=-1$, $f_{id}$ is the center frequency after downconversion and w(n) is environmental clutter and noise.

For a BPSK signal, parameter estimation can be performed with a minimum mean square error estimator
(2)$$(\hat{c_{k}},\hat{f_{id}},\hat{\varphi_{0}})=\underset{\begin{array}{l} c(n)=\{-1,+1\}\\ f\in R\\ \varphi \in [-\pi ,\pi ] \end{array}}{\arg\min}\frac{1}{2}{\left\|c(n)e^{j2\pi fn+\varphi_{0}}-s_{dif}(n)\right\|}_{2}^{2}$$
where $\hat{c_{k}}$, $\hat{f_{id}}$, $\hat{\varphi_{0}}$ and *c*(*n*), f, φ are the estimated value and hypothetical variable of the PN codes, center frequency and initial phase, respectively. In (2), there are three parameters that need to be estimated and they are coupled with each other. It is difficult and complex to solve (2) with a direct searching method. Therefore, we consider two step method which decouples the three parameters by estimating its center frequency and converting it to zero intermediate frequency (ZIF).

## 3. PN Code Estimation Based on NCTVR

In this paper, a novel NCTVR-based PN code estimation method is proposed instead of the minimum mean square error estimator shown in (2) to reduce the complexity of the parameter estimation of a BPSK signal. The proposed method, whose diagram is shown in Figure 2, implements PN code estimation in two steps. First, it pretreats the digital intermediate frequency signal expressed in (1) by estimating its center frequency and convert it to ZIF by secondary downconversion. After pretreatment, the center frequency of the digital intermediate frequency is removed, and the phase jumps is transformed into the level jumps of the ZIF signal. Then, a minimax-concave function-based NCTVR filter is applied to extract the PN codes from the ZIF signal.

### 3.1. Pretreatment

Here, we select an IFC estimator to estimate the center frequency of the digital intermediate frequency signal. To decouple the center frequency and phase jumps of the BPSK, the digital intermediate frequency signal is frequency doubled by subtracting the square of the real part of the digital intermediate frequency signal from the square of its imaginary part. The frequency doubling signal is denoted as
(3)$$\begin{array}{l} s_{FD}(n)=\cos4\pi (2f_{id}n+\varphi_{0})+w_{\mathrm{im}}(n)p(n)\cos2\pi (f_{id}n\\ +\varphi_{0})+w_{\mathrm{re}}(n)p(n)\sin(2\pi f_{id}n+\varphi_{0})+[w_{\mathrm{im}}{}^{2}(n)-w_{\mathrm{re}}{}^{2}(n)]/2 \end{array}$$
where $p(n)=\sum_{k=0}^{K-1}c_{k}rect(n-k\frac{T_{c}}{T_{s}})$, $p(n)\cos2\pi (f_{id}n+\varphi_{0})$ and $p(n)\sin(2\pi f_{id}n+\varphi_{0})$ are the real part and imaginary part of the digital intermediate frequency signal, respectively, and $w_{\mathrm{re}}(t)$ and $w_{\mathrm{im}}(t)$ are the noise accompanied by the real part and imaginary part of the digital intermediate frequency signal, respectively. According to (3), the frequency doubling signal only includes one frequency component $2f_{id}$ and the last three terms are noise. After frequency doubling, an IFCs estimator is adopted to estimate the center frequency. The IFCs frequency estimator, shown in Figure 3., divides frequency estimation into two steps. First, a coarse search is performed based on an FFT, which returns the index of the bin with the largest magnitude. After the coarse search, two DFT coefficients at the bin edges are then calculated and used to interpolate the true center frequency.

Assuming the total number of samples N is an integral power of 2 (This can be achieved by zero padding), the index of the bin with the largest magnitude after FFT is $\hat{m}$. Then, the result of the course search is $\frac{\hat{m}}{N}f_{s}$. Denoting the true frequency of the frequency doubling signal is $\frac{2f_{id}N}{f_{s}}=\hat{m}+\delta$, and |δ|≤0.5 is a residual. The goal of the fine search is to obtain an estimation of *δ*. For this purpose, the two DFT coefficients between $\hat{m}$ are calculated by
(4)$$\begin{array}{l} X(\hat{m}\pm 0.5)=\sum_{n=0}^{N-1}s_{FD}(n)e^{-j2\pi (\hat{m}\pm 0.5)\frac{f_{s}}{N}n}=\sum_{n=0}^{N-1}\cos4\pi (2f_{id}n+\varphi_{0})e^{-j2\pi (\hat{m}\pm 0.5)\frac{f_{s}}{N}n}+W(\hat{m}\pm 0.5)\\ =\sum_{n=0}^{N-1}\cos4\pi (2\frac{\hat{m}}{N}f_{s}n+\delta n+\varphi_{0})e^{-j2\pi (\hat{m}\pm 0.5)\frac{f_{s}}{N}n}+W(\hat{m}\pm 0.5)\\ =\sum_{n=0}^{N-1}e^{j2\varphi_{0}}e^{-j2\pi (\delta \pm 0.5\frac{f_{s}}{N})n}+W(\hat{m}\pm 0.5)=e^{j2\varphi_{0}}\frac{1+e^{j2\pi \delta_{n}}}{1-e^{j2\pi (\frac{\delta_{n}\pm 0.5}{N})}}+W(\hat{m}\pm 0.5) \end{array}$$
where $W\left(\hat{m}\pm 0.5\right)$ is the spectrum of the last three terms in (3). As $\frac{\delta \pm 0.5}{N}\ll 1$, the denominator of (4) can be expanded by the Taylor series as
(5)$$X(\hat{m}\pm 0.5)=b\frac{\delta }{\left(\delta \pm 0.5\right)}+W(\hat{m}\pm 0.5)$$
where $b=\frac{Ne^{j2\varphi_{0}}(1+e^{j2\pi \delta })}{j2\pi \delta }$. When there are no noise terms, the residual can be estimated by
(6)$$\hat{\delta }=\frac{1}{2}\frac{b\frac{\delta }{\left(\delta -0.5\right)}-b\frac{\delta }{\left(\delta +0.5\right)}}{b\frac{\delta }{\left(\delta -0.5\right)}+b\frac{\delta }{\left(\delta +0.5\right)}}=\frac{1}{2}\frac{\left|X(\hat{m}-0.5)\right|-\left|X(\hat{m}+0.5)\right|}{\left|X(\hat{m}-0.5)\right|+\left|X(\hat{m}+0.5)\right|}$$

The noise terms will affect the accuracy of the residual, which is analyzed later. Finally, the estimated result of center frequency can be expressed as
(7)$$\hat{f_{id}}=\frac{1}{2}\frac{\hat{m}}{N}f_{s}+\frac{1}{4}\frac{\left|X(\hat{m}-0.5)\right|-\left|X(\hat{m}+0.5)\right|}{\left|X(\hat{m}-0.5)\right|+\left|X(\hat{m}+0.5)\right|})$$

Its estimation accuracy can be improved by updating the value of $\hat{m}$ with $\hat{m}=\hat{m}+\hat{\delta }$ and repeating two search steps. The estimation accuracy is 1.0147 times the asymptotic Cramer–Rao bound.

The initial phase of the frequency doubling signal is estimated by the Fourier coefficients at $\pm (\hat{m}+\hat{\delta })$. The DFT coefficients of the frequency doubling signal at $\pm (\hat{m}+\hat{\delta })$ can be calculated as
(8)$$X[\pm (\hat{m}+\hat{\delta })]=\sum_{n=0}^{N-1}s_{FD}(n)e^{\pm j2\pi (\hat{m}+\hat{\delta })\frac{f_{s}}{N}n}=e^{\pm 2\varphi_{0}}|X[(\hat{m}+\hat{\delta })]|$$

Namely, $e^{\pm \varphi_{0}}=X[\pm (\hat{m}+\hat{\delta })]/|X[(\hat{m}+\hat{\delta })]|$. According to Euler formula, we can obtain
(9)$$\sin2\varphi_{0}=\frac{e^{2\varphi_{0}}-e^{-2\varphi_{0}}}{2j}\;\cos2\varphi_{0}=\frac{e^{2\varphi_{0}}+e^{-2\varphi_{0}}}{2}$$

According to (9), the estimated result of the initial phase $\hat{\varphi_{0}}$ can be obtained. However, the frequency doubling resulting in the solution is not unique. The real value of the $\varphi_{0}$ maybe $\hat{\varphi_{0}}$ or $\hat{\varphi_{0}}+\pi$. To solve this problem, a posterior, the correlation ZIF signal and the original signal, is adopted. When their correlation is positive, $\varphi_{0}=\hat{\varphi_{0}}$; otherwise $\varphi_{0}=\hat{\varphi_{0}}+\pi$.

The final ZIF signal obtained after secondary downconversion can be expressed as
(10)$$\begin{array}{ll} s_{\mathrm{ZIF}}(n) & =[\sum_{k=0}^{K-1}c_{k}rect(n-kT_{c})e^{j2\pi f_{id}n+\varphi_{0}}+w(n)]e^{-j2\pi \hat{(f_{id}}n+\hat{\varphi_{0}})}\\ & =\sum_{k=0}^{K-1}c_{k}rect(n-kT_{c})e^{j2\pi (f_{id}-\hat{f_{id}})n+(\varphi_{0}-\hat{\varphi_{0}})}+w(n)e^{-j2\pi \hat{(f_{id}}n+\hat{\varphi_{0}})} \end{array}$$

### 3.2. PN Code Estimation Based on NCTVR

After secondary downconversion, the ZIF signal is sent to the NCTVR filter. To facilitate the analysis, we assume that $f_{id}=\hat{f_{id}}$ and $\varphi_{0}=\hat{\varphi_{0}}$. In fact, there are always errors between the measured values of frequency and phase and their real values. However, as $f_{id}-\hat{f_{id}}\ll \frac{1}{KT_{c}}$ with the IFC estimator, the frequency residue $f_{id}-\hat{f_{id}}$ has little effect on the polarity of the PN code. The same is true for the participation of the phases $\varphi_{0}-\hat{\varphi_{0}}$. Under this assumption, (10) can be rewritten as
(11)$$s_{\mathrm{ZIF}}(n)\;=\sum_{k=0}^{K-1}c_{k}rect(n-kT_{c})+w(n)e^{-j2\pi \hat{(f_{\mathrm{id}}}n+\hat{\varphi_{0}})}$$

Then, PN code estimation can be formulated as the following optimization problem:(12)$$\underset{c\in {\left\{-1,1\right\}}^{N},}{\hat{c}=\arg\min}\frac{1}{2}{\left\|s_{Zif}-c\right\|}_{2}^{2},$$
where $\hat{c}$ is the estimated codes. As the nonpositive definition of (12), a minimax-concave penalty regularization function is adopted as a Lagrangian penalty factor and the following optimization objectives can be obtained,
(13)$$\hat{c}=\underset{c\in {\left\{-1,1\right\}}^{N}}{\arg\min}\frac{1}{2}{\left\|s_{Zif}\;-c\right\|}_{2}^{2}+\lambda \sum_{i=1}^{N}\phi (\left\|{\left[Dc\right]}_{i}\right\|;a)$$
where D represents the first-order derivatives padded by Neumann boundary conditions and ϕ(t,a) is the minimax-concave penalty function, which is defined by
(14)$$\phi (t,a)=\{\begin{array}{c} -\frac{a}{2}t^{2}+\sqrt{2a}t,t\in [0,\sqrt{2/a})\\ 1,t\in [\sqrt{2/a},+\infty ) \end{array}$$
where a in ϕ(t,a) affects the degree of nonconvexity. When a→∞, the MC penalty tends to be L0 norm. For a=0, the ϕ(t,a) is defined as ϕ(t,a)=|t|, namely the MC penalty tends to be L1 norm. The curve of ϕ(t,a) with different a is shown in Figure 4.

Compared with the L1 norm, the proposed MC penalty promotes sparsity more effectively and accurately preserves the amplitude of the piecewise-constant signal. Shown in Figure 5a, the amplitudes of the estimation results from MC penalty are ±1, which are same as the real PN codes. However, the amplitudes of the estimation results from L1 norm tend to be random. Therefore, the solution of (14) can be imposed a constraint to force $\hat{c}$ to be a binary vector. However, when adopting L1 norm to estimate the PN codes, we need to adopt additional binary quantization algorithms. The MC penalty possesses a better noise tolerance than the L1 norm. Shown in Figure 5b, many errors have occurred in the results of the L1 norm with SNR = 0 dB.

When $0\leq a\leq \frac{1}{4\lambda }$, the cost function is strongly convex, and (13) can be rewritten as
(15)$$\hat{c}=\underset{c\in {\left\{-1,1\right\}}^{N}}{\arg\min}\frac{1}{2}{\left\|s_{Zif}\;-c\right\|}_{2}^{2}+\lambda {\left\|Dc\right\|}_{1}-\lambda \underset{v\in R^{N}}{\min}\{{\left\|v\right\|}_{1}+\frac{a}{2}{\left\|Dc-v\right\|}_{2}^{2}\}$$
where, $v\in R^{N}$ is an intermediate variable. As the discontinuous minimax-concave penalty function, the extremum of (15) cannot be solved with its first-order or second-order derivative. Here, we propose an iterative algorithm based on forward–backward splitting algorithm. Defining the two functions
(16)$$\begin{array}{l} f(c)=\underset{c\in {\left\{-1,1\right\}}^{N}}{\arg\min}\frac{1}{2}{\left\|s_{Zif}\;-c\right\|}_{2}^{2}-\lambda \underset{v\in R^{N}}{\min}\{{\left\|v\right\|}_{1}+\frac{a}{2}{\left\|Dc-v\right\|}_{2}^{2}\}\\ g(c)=\lambda {\left\|Dc\right\|}_{1} \end{array}$$

According to the above definitions, f(c) and g(c) are convex, and $\frac{\partial f}{\partial c}$ is Lipschitz continuous. Hence, the minimization of (15) can be solved based on the forward–backward splitting algorithm for minimization by iterating
(17)$$\begin{array}{l} z^{p}=c^{p}-u\frac{\partial }{\partial c^{p}}f\\ c^{p+1}=\underset{v}{\arg\min}\{\frac{1}{2}{\left\|z-v\right\|}_{2}^{2}+\mu g(v)\} \end{array}$$
where p is the number of iterations and $\frac{\partial f}{\partial c}$ represents the first-order derivative. Taking μ=1, (17) can be expressed as
(18)$$z^{p}=aD^{T}(Dc^{p}-soft_{1/a}(Dc^{p}))$$
(19)$$c^{p+1}=\underset{v\in {\left\{-1,1\right\}}^{N}}{\arg\min}\{\frac{1}{2}{\left\|s_{\mathrm{Zif}}+\lambda z^{p}-v\right\|}_{2}^{2}+\lambda {\left\|Dc\right\|}_{1}\}$$
where $soft_{1/a}(t)=\{\begin{array}{c} 0,\left|y\right|<1/a\\ (\left|t\right|-\lambda )sign(t),\left|y\right|>1/a \end{array}$ is a soft threshold function. We observe that the backward step in (19) is a standard one-dimensional TVR problem, and the intermediate variables **z** play a role in enhancing sparsity. The extra computational complexity of MC penalty is (18), which includes twice matrix multiplications and a soft threshold function. The flow chart of NCTVR is shown in Figure 6.

### 3.3. The Motivation behind the NCTVR

The essence of the PN code estimation method proposed in this paper is to extract level jumps from a noise-polluted ZIF signal. According to (18) and (19), when **z** is equal to zero, (19) is a classical TVR problem. The classical TVR problem can be solved with an L1-norm-based regularization penalty term, but it has a limitation that tends to underestimate the amplitudes of signal discontinuities. Therefore, we introduce a minimax-concave penalty function to improve the TVR problem. The sequence **z**, resulting from the minimax-concave penalty function, enhances the sparsity of the ZIF signal and makes it more robust to noise.

As observed in Figure 7., sequence z (the red line in Figure 7.), which is computed by (18) upon convergence of the NCTVR, acts as a detector of polarity change. By applying the soft threshold function in (20), we obtain
(20)$$z(n)=\{\begin{array}{c} D^{T}Dc(n),\;\left|Dc(n)\right|\;>\frac{1}{a}\\ 0,\;\left|Dc(n)\right|\;>\frac{1}{a} \end{array}$$

Therefore, the sequence **z** can be considered as the response of a Laplacian operator (LAPO) on the PN codes c. The Laplacian operator takes the second derivative of the ZIF signal: When the PN codes do not change, the Laplacian operator outputs zero, corresponding to jumps with a minimal jump height lower than 1/a caused by noise. As illustrated in Figure 7b, if PN codes change from −1 to +1 (|Dc(n)|>0), z shows a positive impact first and then a negative impact.

## 4. Performance of NCTVR

### 4.1. Variance of the Estimated Centre Frequency

Assume that w(n) is zero-mean additive Gaussian noise with variance $\sigma^{2}$. According to (6), the noise carried by the frequency doubling signal can be expressed as
(21)$$w_{FD}(n)=w_{im}(n)\cos2\pi (f_{id}n+\varphi_{0})+w_{re}(n)\sin(2\pi f_{id}n+\varphi_{0})+[w_{im}{}^{2}(n)-w_{re}{}^{2}(n)]/2$$

As the last item of (21) is a higher order infinitesimal of variance $O\left(\sigma^{2}\right)$, $w_{FD}(n)$ can still be considered zero-mean additive Gaussian noise with the same variance $\sigma^{2}$. Therefore, the variance of the estimated center frequency can be expressed as
(22)$$Var(\hat{f_{id}})=\frac{f_{s}^{2}\pi^{2}{\left(\delta -0.25\right)}^{2}(4\delta^{2}+1)}{4N^{3}\rho \cos^{2}(\pi \delta )}$$
where ρ is the SNR. By iteration, it has a minimum
(23)$$Var(\hat{f_{id}})=\frac{f_{s}^{2}\pi^{2}}{64N^{3}\rho }$$

As our initial phase estimation method is still based on Fourier coefficients, it variance of the estimated center frequency can be expressed as
(24)$$Var(\hat{\varphi_{0}})=\frac{\pi \sigma }{2N|X[+(\hat{m}+\hat{\delta })]|}+\frac{\pi \sigma }{2N|X[-(\hat{m}+\hat{\delta })]|}$$

### 4.2. Accuracy of The Estimated PN Codes

According to (18) and (19), NCTVR detects the PN codes with the first and second derivatives of the ZIF signal. The detection threshold is decided by the two regularization parameters λ and a. When |Dc|>λ and $\left|D^{T}Dc\right|>\frac{1}{a}$, it is considered to be a polarity change. Therefore, the regularization parameters are set between the derivative of the signal and noise. When the SNR is low, the derivative of the noise may be larger than that of the PN codes, and some estimation errors appear.

## 5. Simulations and Experiments

### 5.1. Simulations

The performance of the NCTVR-based PN codes estimation method is tested by numerical simulation and semiphysical in this section. In our simulation scenario, there is only one BPSK radar that works in the X band with a carrier frequency of 10 GHz. Its chipping width and code length are 50 ns and 31, respectively. The reconnaissance receiver, working under the SNR varying from −10~10 dB, is able to accept signals ranging from 10 GHz to 11 GHz by using a local oscillator of 10.5 GHz and high-speed ADCs with a 1 GHz sampling rate. As the distance between the radar and reconnaissance receiver changes slowly in a short time, the amplitude change of the received signal can be ignored. When the BPSK signal enters the reconnaissance receiver, it is quadrature downconverted and sampled to a digital intermediate frequency signal. When processing the obtained digital intermediate frequency signal, the iterations of the IFCs and NCTVR are 2 and 100, respectively, with regularization parameters where λ=0.5 and a=1/4λ.

The effect of the pretreatment is estimating the center frequency and initial phase of the digital intermediate frequency signal and converting it to ZIF. The standard deviation of the center frequency and initial phase under different SNRs is shown in Figure 8. According to the result shown in Figure 8a, the IFCs estimator possesses an extremely high frequency estimation accuracy, and its standard deviation approaches the asymptotic Cramer–Rao bound, which is lower than 20 kHz with N = 1550. According to the result shown in Figure 8b, the standard deviation of the initial phase is lower than 0.25 rad or 15° even under SNR = −10 dB. Assuming the frequency estimation error is 20 kHz and the initial phase estimation error is 0.25 rad, the ZIF signal after downconversion, which is a level-jumping signal that is polluted by noise, is shown in Figure 8c. Namely, the pretreatment can effectively transform phase jumps of the BPSK signal into the level jumps of the ZIF signal.

The simulation results of the NCTVR with different SNRs are shown in Figure 9. From Figure 9a–c, the estimation results are almost identical to the original sequence with an SNR larger than 0 dB. Only 6 of the 1550 sampling points are estimated incorrectly when the SNR is −5 dB and 174 of the 1550 sampling points are estimated incorrectly when the SNR is −10 dB, which are shown in Figure 9d,e. According to the simulation results, the NCTVR can efficiently extract the PN codes from noise polluted ZIF signals even under serious electromagnetic environments. When the input SNR is higher than −5 dB, its statistical miscalculation is lower than 5% of the total sample number. However, when the input SNR is lower than −5 dB, its miscalculation increases rapidly over 10%. This is because the probability that the derivative of the noise is larger than that of the PN codes is greatly improved when the SNR deteriorates from −5 dB to −10 dB. 

Estimation performance of the MC penalty and L1 norm is shown in Figure 10. When SNR = 0 dB, the estimation results of the MC penalty is almost the same as the original codes. However, there are many estimation errors when adopting L1 norm.

To more intuitively reflect the estimation accuracy of the NCTVR, 5000 simulations under different SNRs are carried out, and the average correlation coefficients between the estimation results and the original PN codes are selected as the basis for evaluation. From the simulation results shown in Figure 11, the estimation accuracy of NCTVR (blue line) is over 95% when SNR is higher than 0 dB. As SNR continues to deteriorate, more estimation errors appear, and its estimation accuracy decreases to 0. 70 when SNR = −10 dB. Namely, the NCTVR has an extremely high estimation accuracy when the SNR is larger than 0 dB. When SNR is lower than −5 dB, the estimation accuracy deteriorates rapidly. This is because the NCTVR mainly attempts to detect the amplitude of the LAPO of the ZIF signal; With SNR lower than −5 dB, the LAPO of the PN codes is covered by that of the noise. Compared with the L1 norm (purple line), the MC penalty shows a higher estimation accuracy for that the MC penalty possesses a better noise tolerance than L1 norm. Compared with the method proposed in [18] (yellow line), the NCTVR-based method possesses a higher estimation accuracy. The accuracy of our method is a little lower than the method proposed in [16] when SNR is lower than 0 dB. It is because the duffing oscillator is sensitive to periodic signal and insensitive to noise. However, [16] can only detects the polarity changes of the PN codes and needs the starting symbol of the PN code as a priori to make sure the polarity of the following code. This makes it hard to be realized in application.

### 5.2. Experiments

To verify the performance of NCTVR, semiphysical tests are carried out in an anechoic chamber. In our experiment, the BPSK radar is 30 m away from the reconnaissance receiver. Its transmitting signal, with a 10 GHz carrier frequency and a chipping width of 20 ns, is captured and downconverted by an N8201A, and the obtained intermediate frequency signal, whose center frequency is approximately 100 MHz, is sampled by an M9203A digital receiver with a sampling rate of fs = 1 GHz. The final digital intermediate frequency signal is processed on a PC. 

The results of the NCTVR are shown in Figure 12. Figure 12a shows the captured signal from the time domain (yellow line) and frequency domain (red line). The bandwidth of the captured signal is 50 MHz. The ZIF signal is shown in Figure 12b, from which we conclude that pretreatment leads the BPSK signal to ZIF and transforms the phase jumps into the level jumps of ZIF. The result of the NCTVR is shown in Figure 12c, from which we see that the NCTVR can extract the PN codes from the ZIF signal. According to the results of the experiments, the NCTVR-based method proposed in this paper works well with the experimental data.

## 6. Conclusions

In this paper, a novel PN codes estimation method based on NCTVR is proposed for radar jammer. the NCTVR-based method first converts the digital intermediate frequency signal into ZIF with a pretreatment based on IFCs and secondary downconversion. This turns phase jumps of the BPSK signal into level jumps of the ZIF signal. Then, an MC penalty based NCTVR filter is introduced to extract level jumps of the ZIF signal. By adopting the MC penalty function instead of the L1-norm-based penalty, the NCTVR can extract accurately the level jumps of the ZIF signal under a serious SNR. Its estimation accuracy is over 70%, even under SNR −10 dB. Compared with the existing PN code estimation method, the advantage of the proposed algorithm is that it possesses a competitive estimation accuracy and do not need any priories. The results of the simulations and semiphysical tests show that the proposed method works well on a radar jammer.

However, the proposed method needs an iterative algorithm to solve the optimistic problem, which affects its real-time performance. Therefore, our future work includes solving the NCTVR by ADMM and adopting the deep-ADMM-net to accelerate its speed.

## Figures and Tables

**Figure 1 sensors-23-00554-f001:**
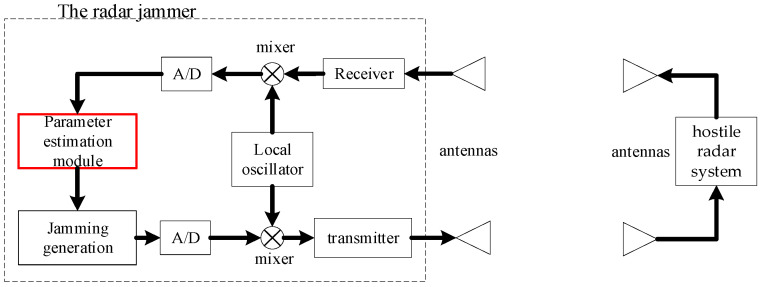
Principle of the radar jammer.

**Figure 2 sensors-23-00554-f002:**
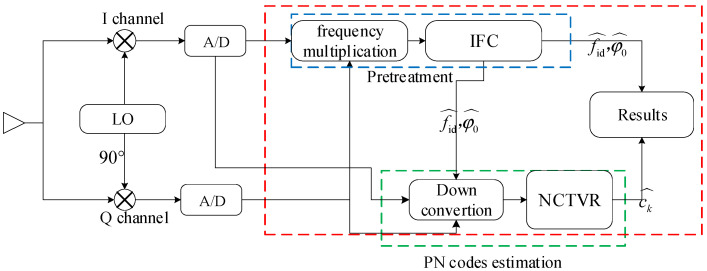
Diagram of the NCTVR-based method. LO: local oscillator.

**Figure 3 sensors-23-00554-f003:**
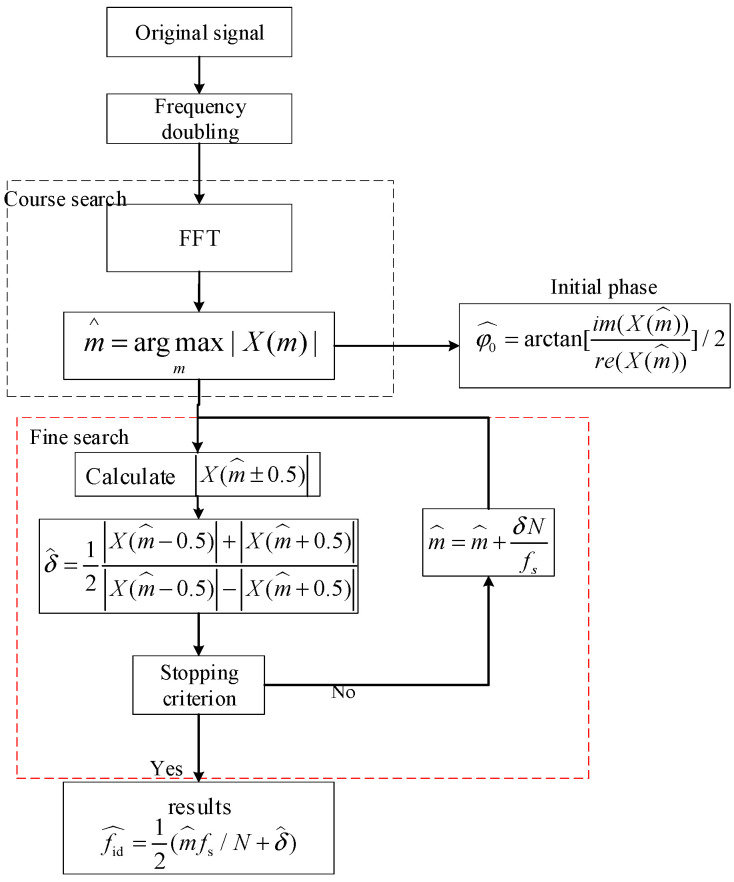
Flow chart of center frequency estimation.

**Figure 4 sensors-23-00554-f004:**
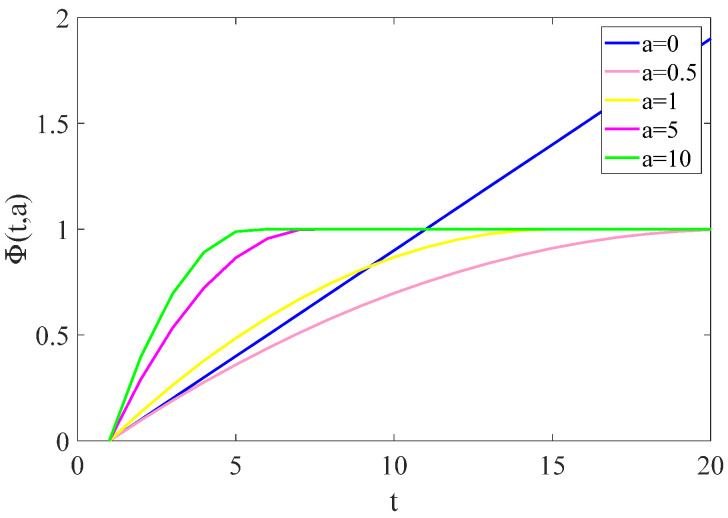
The curve of ϕ(t,a).

**Figure 5 sensors-23-00554-f005:**
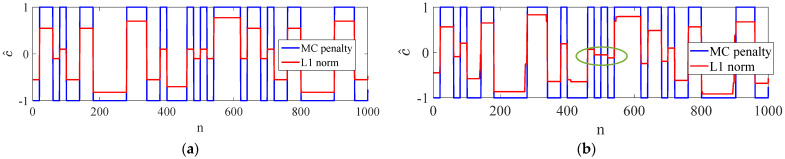
Performance of the MC penalty and L1 norm, (**a**) without noise, (**b**)with SNR = −10 dB.

**Figure 6 sensors-23-00554-f006:**
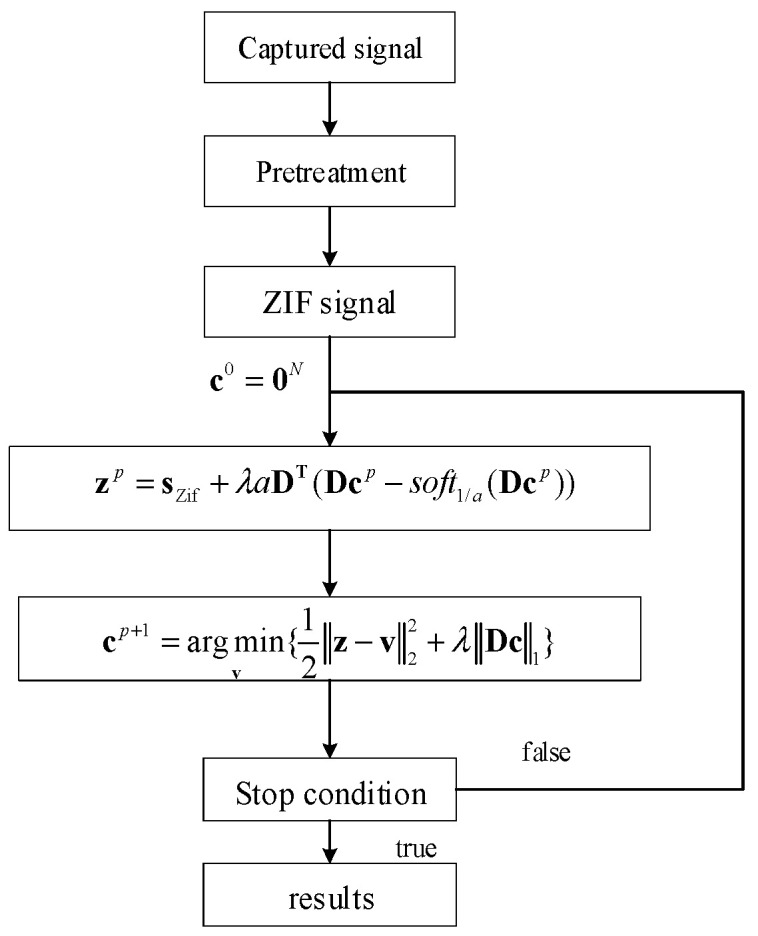
Flow chart of the NCTVR.

**Figure 7 sensors-23-00554-f007:**
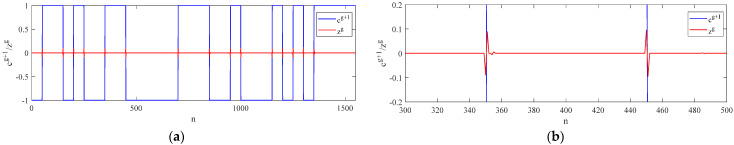
The response of the NCTVR: (**a**) global and (**b**) local.

**Figure 8 sensors-23-00554-f008:**
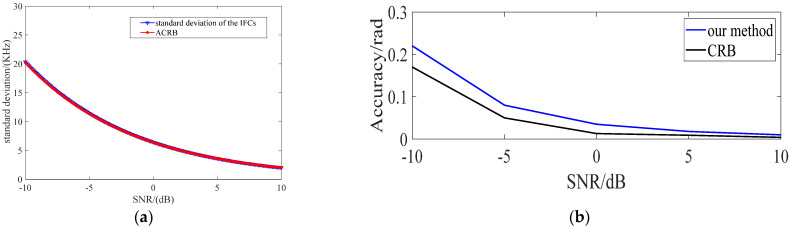
Results of the pretreatment: (**a**) standard deviations of estimated frequency; (**b**) standard deviations of estimated initial phase; (**c**) the ZIF signal after pretreatment under SNR = −5 dB.

**Figure 9 sensors-23-00554-f009:**
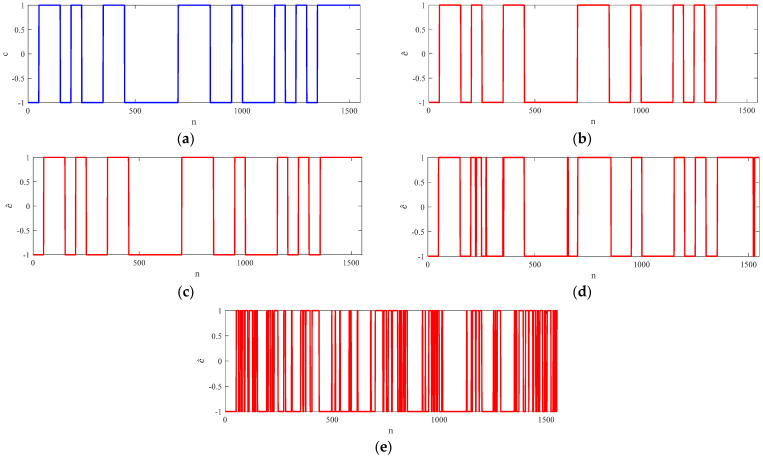
The estimated PN codes under different SNRs. (**a**) Original sequences; (**b**) SNR = 5 dB; (**c**) SNR = 0 dB; (**d**) SNR = −5 dB; (**e**) SNR = −10.

**Figure 10 sensors-23-00554-f010:**
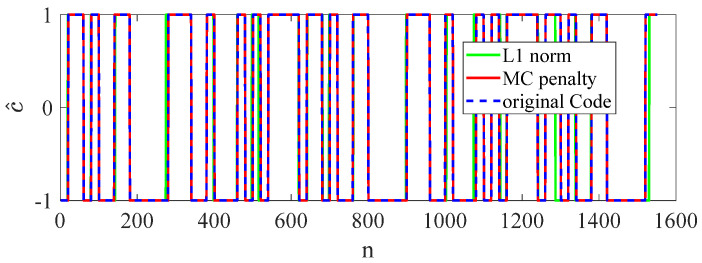
Estimation performance of the MC penalty with SNR = 0 dB.

**Figure 11 sensors-23-00554-f011:**
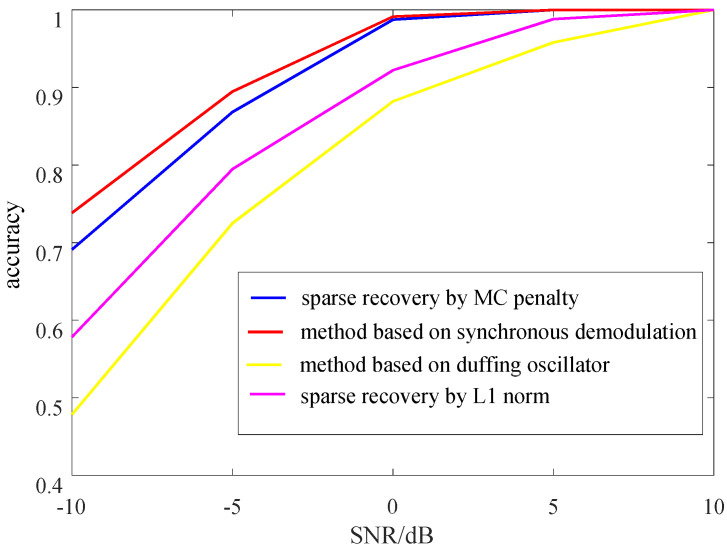
Correlation coefficients under different SNRs.

**Figure 12 sensors-23-00554-f012:**
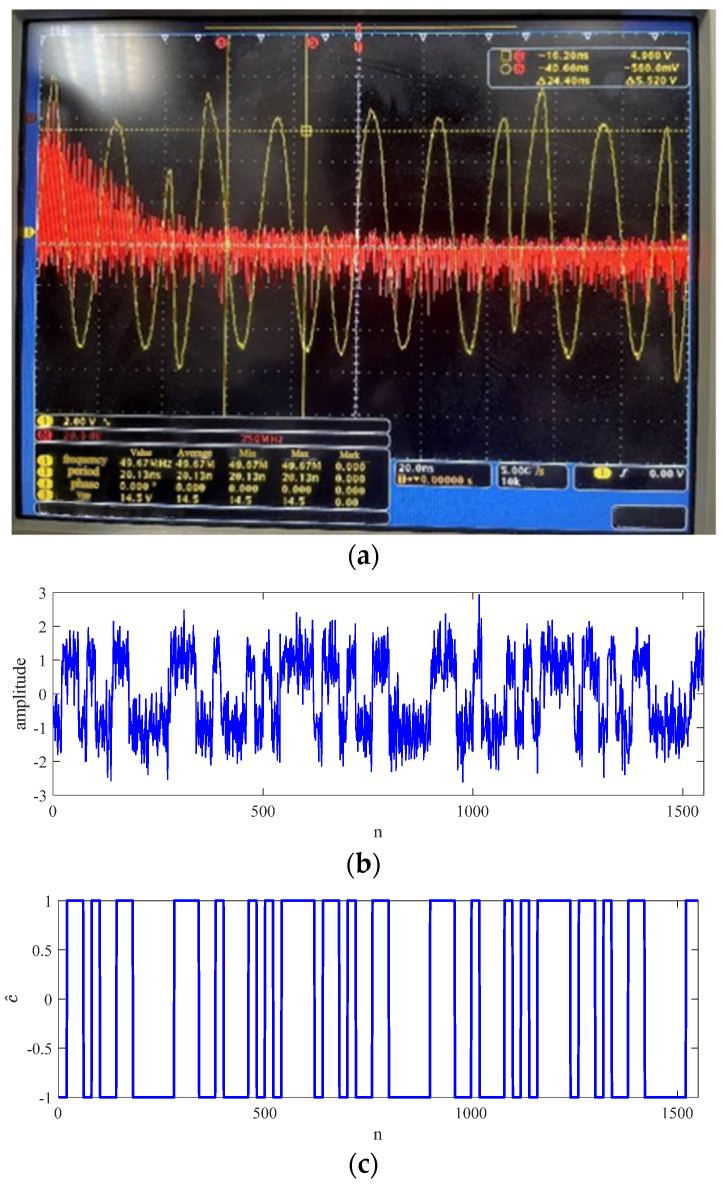
Results of the semiphysical tests. (**a**) The captured BPSK signal in the time domain and frequency domain. (**b**) The ZIF signal obtained from pretreatment. (**c**) The estimated PN codes.

**Table 1 sensors-23-00554-t001:** Summary of existing methods.

Existing Methods	Principle	Limitations
Methods proposed in [10,11,12] and [18,19,20,21]	They adopt time frequency analysis to estimate the chip rate and carrier frequency of the BPSK signal	They are unable to estimate the PN codes.
Two-stage method proposed in [24]	It adopts the cross correlation to estimate the PN codes of the BPSK signal in serious SNR.	It is only suitable for Barker codes 7, 11 and 13.
Method proposed in [23]	It adopts matrix eigen decomposition to estimate the PN codes of the BPSK signal.	It needs to know the chip rate and period of the PN code as a priori
Method proposed in [16]	It uses the state changes of the duffing oscillator to estimate the PN codes of the BPSK signal in serious SNR	It only detects the polarity changes of the PN codes and needs to know the polarity of starting symbol as a priori.
Our two-stage method.	It uses the sparsity of the PN codes in time domain to estimate the PN codes of the BPSK signal in serious SNR	\

## Data Availability

Not applicable.

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
