# Peer review of "PN Codes Estimation of Binary Phase Shift Keying Signal Based on Sparse Recovery for Radar Jammer"

_sensors, 2023, doi:10.3390/s23010554_

Round 1

Reviewer 1 Report

This paper proposed two-step pseudorandom code estimation method of binary phase shift keying (BPSK) signals based on sparse recovery. numeric simulations and semi physical tests were used to verify the effectiveness of the proposed method. The result indicated that the proposed method is able to estimate pseudorandom codes from the captured BPSK signal in serious electromagnetic environments. Overall, the study is interesting, but it current form needed more further enhancements, such as:

- a new literature review/related studies section should be provided by presenting the previous related studies to BPSK signals estimation. A table summarizing all the previous studies and contrast with your current work should also be provided to verify and justify the unique contributions of your work.

- In figure 7, please elaborate why your method has lower accuracy than the method proposed in [16].

- The proposed method is interesting, but it’s hardly found its usefulness in term of sensors journal reader. Please provide the practicability/ expected scenarios/applications of your method in real-world.

- The limitations and future works should be elaborated in the conclusions section.

- Most of the references used are outdated. Please add works from recent years, such as 2020—2022 whenever possible. 

Author Response

Thank you very much for comments and professional advices on our manuscript. Based on your suggestions and requests, we have made corrected modifications on the revised manuscript and the changes are highlighted with yellow in the revised manuscript. We hope that our work can be improved again. Furthermore, the details of the modifications are shown as follows.

Reviewer 2 Report

In this article, the authors discussed a method of estimating the parameters of BPSK signals, or extracting the BPSK code. Extracting pseudorandom codes is crucial in radar, communication models. In this field it is known that extracting pseudorandom codes is a complex problem.

Here, authors proposed a novel two step pseudorandom code estimation method. This method first estimates the center frequency with an IFC frequency estimator and converts it to ZIF, then uses an iterative algorithm based on the forward-backward splitting algorithm to search for the minimum of the NCTVR optimization problem

 But there are still some problems needs to be further explained.

1      I think that the authors should give more details on the difference between L1 norm and minimax-concave function, especially on the calculate load and optimize performance;

2      I think the authors should describe the accurate phase estimation method rather than the coarse phase estimation and how to solve the 180 problem (the phase reversal problem).

3   As a general comment, I would recommend to carefully review the use of English, and punctuation of the text.

Author Response

Thank you very much for comments and professional advices on our manuscript. Based on your suggestions and requests, we have made corrected modifications on the revised manuscript and the changes are highlighted in the revised manuscript. We hope that our work can be improved again. Furthermore, the details of the modifications are shown as follows.

Reviewer 3 Report

This paper proposes a novel pseudorandom code estimation method based on zero intermediate frequency conversion and convex optimization processing, and the method has been verified by simulation and real data experiments. However, there are still some problems need improvement:

1. The Ck is not clear in Figure 1, and it should be revised.

2. There are two fid and pha0 in Line 170, 174, 176, and these two statements are indistinct. Please give the accurate definition of these statements.

3. The vector v in Equation 14 has not definition, therefore the meaning of the vector v is not clear.

4. I don’t understand the application scenario of the proposed method, passive radar or signal detection?

5. The signal estimation performance comparison is not sufficient, namely the proposed method should be compared with some classic methods.

Author Response

(The authors gave the same response as above.)

Round 2

Reviewer 2 Report

I think now it can be published